# Prevalence and Characteristics of Plasmid-Encoded Serine Protease EspP in Clinical Shiga Toxin-Producing *Escherichia coli* Strains from Patients in Sweden

**DOI:** 10.3390/microorganisms12030589

**Published:** 2024-03-15

**Authors:** Lei Wang, Ying Hua, Xiangning Bai, Ji Zhang, Sara Mernelius, Milan Chromek, Anne Frykman, Sverker Hansson, Andreas Matussek

**Affiliations:** 1Department of Microbiology, Division of Laboratory Medicine, Institute of Clinical Medicine, University of Oslo, 0372 Oslo, Norway; lei.wang@medisin.uio.no (L.W.); hying615@smu.edu.cn (Y.H.); 2Jinan Center for Disease Control and Prevention, Jinan 250021, China; 3Department of Microbiology, Division of Laboratory Medicine, Oslo University Hospital, 0372 Oslo, Norway; xiangning.bai@medisin.uio.no; 4Department of Clinical Microbiology, Division of Laboratory Medicine, Karolinska Institutet, 141 52 Stockholm, Sweden; 5Fonterra Research and Development Centre, Dairy Farm Road, Palmerston North 4442, New Zealand; ji.zhang@fonterra.com; 6Laboratory Medicine, Department of Clinical and Experimental Medicine, Linköping University, Jönköping Region County, SE-581 83 Linköping, Sweden; sara.mernelius@rjl.se; 7Division of Pediatrics, Department of Clinical Science, Intervention and Technology, Karolinska Institutet, Karolinska University Hospital, 141 52 Stockholm, Sweden; milan.chromek@regionstockholm.se; 8Department of Pediatrics, Institute of Clinical Sciences, Sahlgrenska Academy, University of Gothenburg, 40530 Gothenburg, Sweden; anne.frykman@regionvarmland.se (A.F.); sverker.hansson@pediat.gu.se (S.H.); 9Queen Silvia Children’s Hospital, Sahlgrenska University Hospital, 416 50 Gothenburg, Sweden

**Keywords:** Shiga toxin-producing *Escherichia coli*, *espP* gene, gene diversity, hemolytic uremic syndrome, clinical significance

## Abstract

Shiga toxin-producing *Escherichia coli* (STEC) infection can cause a broad spectrum of symptoms spanning from asymptomatic shedding to mild and bloody diarrhea (BD) and even life-threatening hemolytic-uremic syndrome (HUS). As a member of the serine protease autotransporters of *Enterobacteriaceae* (SPATE) family, EspP has the ability to degrade human coagulation factor V, leading to mucosal bleeding, and also plays a role in bacteria adhesion to the surface of host cells. Here, we investigated the prevalence and genetic diversity of *espP* among clinical STEC isolates from patients with mild diarrhea, BD, and HUS, as well as from asymptomatic individuals, and assessed the presence of *espP* and its subtypes in correlation to disease severity. We found that 130 out of 239 (54.4%) clinical STEC strains were *espP* positive, and the presence of *espP* was significantly associated with BD, HUS, and O157:H7 serotype. Eighteen unique *espP* genotypes (GTs) were identified and categorized into four *espP* subtypes, i.e., *espP*α (119, 91.5%), *espP*γ (5, 3.8%), *espP*δ (4, 3.1%), and *espP*ε (2, 1.5%). *espP*α was widely distributed, especially in strains from patients with BD and HUS, and correlated with serotype O157:H7. Serogroup O26, O145, O121, and O103 strains carried *espP*α only. Ten GTs were identified in *espP*α, and *espP*α/GT2 was significantly associated with severe disease, i.e., BD and HUS. Additionally, *espP* was strongly linked to the presence of *eae* gene, and the coexistence of *espP*α and *stx2*/*stx2a* + *stx2c* was closely related to HUS status. To sum up, our data demonstrated a high prevalence and genetic diversity of the *espP* gene in clinical STEC strains in Sweden and revealed an association between the presence of *espP*, *espP* subtypes, and disease severity. *espP*, particularly the *espP*α subtype, was prone to be present in more virulent STEC strains, e.g., “top-six” serotypes strains.

## 1. Introduction

Shiga toxin-producing *Escherichia coli* (STEC) is a foodborne, gram-negative bacterium belonging to the *Enterobacteriaceae* family and can cause a variety of human diseases ranging from asymptomatic shedding to mild/bloody diarrhea (BD) or even life-threatening diseases such as hemolytic uremic syndrome (HUS) [1]. STEC infection is one of the leading causes of acute kidney injury in children, and STEC-infected individuals aged over 60 are more prone to mortality, irrespective of clinical conditions [2,3]. Although O157:H7 has been considered the top causative serotype of STEC-linked disease and outbreaks, non-O157 strains with various genetic backgrounds are increasingly recognized by their association with HUS and linkage to large outbreaks, particularly strains of the “top-six” serogroups (i.e., O26, O45, O103, O111, O121 and O145) [4,5,6,7]. Shiga toxin (Stx) is the most important virulence factor in STEC. It contains two main types, assigned Stx1 and Stx2, with four Stx1 subtypes (a, c, d, and e) and twelve Stx2 subtypes (a–l) [8]. Stx2 is more critical than Stx1 in the development of HUS [9,10], and strains carrying *stx2a* with/without *stx2c* genes are significantly associated with severe clinical diseases [11]. Intimin, encoded by *eae* gene located within the locus of enterocyte effacement (LEE) pathogenicity island, is an important aggravating factor involved in gut colonization of STEC. Intimin can induce attaching and effacing (A/E) lesions on intestinal epithelial cells and contribute to human diseases, including the development of hemorrhagic colitis (HC) and HUS [12,13].

STEC induces intestinal impairment through the release of virulence factors without invading tissues [14,15]. The release of secreted proteins, such as proteases, is crucial for the generation of A/E lesions and is involved in a variety of processes associated with infection [16]. Extracellular serine protease P (EspP) is one of the most abundant proteins in culture supernatants of STEC strains and has been described as a member of the serine protease autotransporter of *Enterobacteriaceae* (SPATEs) protein family encoded on large virulence plasmids, such as pO157, pO113, and pO26-Vir in STEC strains [17,18,19,20]. The *espP* gene comprises a 3900 bp open reading frame encoding the 1300 amino acid (aa) EspP protein with a molecular weight of 142 kDa, and the mature secreted passenger domain with a molecular weight of 104 kDa is generated through cleavage of the N-terminal signal peptide and the C-terminal β-domain and secreted into the extracellular milieu, showing serine protease activity [21]. By cleaving coagulation factor V and complement C3, C3b, and C5, EspP could impact host proteins, which are important for coagulation and complement activation, thus enhancing the severity of infections [22,23]. EspP might also be involved in the regulation of virulence, as shown by the cleavage of hemolysin [21]. A recent study reported that pooled immunoglobulins (IgG) on the course of disease in a mouse model could bind to EspP, block its enzymatic activity, and protect the host from O157:H7 STEC infection [24]. Additionally, EspP could stimulate electrogenic ion transport in human colonic monolayers, leading to watery diarrhea that is often followed by HC and extra-intestinal complications, including HUS, while neither Stx nor numerous components of the type-III secretion system have been found to independently elicit fluid secretion [25]. Collectively, EspP could promote colonic cell injury, bacterial adherence to intestinal cells, and the uptake of Stx by intestinal cells [25,26], and its role in blood coagulation, pathophysiology, and immune-modulation can contribute to STEC pathogenesis [27,28]. Five EspP subtypes have currently been identified (EspPα-EspPε) [29]. EspPα participates in biofilm formation and also plays a role in adhesive and cytopathic effects [23,30,31]. EspPγ is able to cleave pepsin and human coagulation factor V, while EspPβ and EspPδ either remained un-secreted or exhibited proteolytic activity [22]. Intriguingly, EspPα has been shown to be more prevalent in human isolates, while other *espP* subtypes are more prevalent in reservoir animals and the environment [22,28]. 

Although EspP is frequently found in STEC strains [27], the role of EspP in STEC pathogenesis is not well-studied, and the molecular characteristics of *espP*-positive STEC strains, especially clinical strains, have rarely been described. Therefore, in this study, we investigate the prevalence of *espP* and its subtypes and polymorphisms among clinical STEC strains isolated from patients with varying disease outcomes in Sweden. Furthermore, we assess its correlations with serotypes, other virulence factors such as *eae* and *stx,* and clinical outcomes.

## 2. Materials and Methods

### 2.1. Ethic Statement

The study was approved by both the regional ethics committees in Gothenburg (2015/335-15) and Stockholm (2020-02338), Sweden. Patient consent was waived due to a retrospective review of the patients’ medical records. Patient data were anonymous, and no consent was required to work with the bacterial strains. 

### 2.2. Bacterial Strains

A total of 239 STEC strains were included in this study. These strains were isolated from STEC-infected individuals in Sweden in the period of 1994–2018. The isolation of STEC strains was performed as described previously [32]. Clinical data of STEC-infected patients, such as age, sex, and clinical symptoms, were collected by reviewing medical records and utilizing the standard practices employed for STEC surveillance in Sweden, with clinical symptoms categorized into non-bloody stool (NBS), bloody diarrhea (BD) and HUS. The duration of bacterial shedding was defined as described previously [33]. 

Bacterial DNA of all STEC strains were extracted and then subjected to whole-genome sequencing using Illumina HiSeq X platform at SciLifeLab (Stockholm, Sweden) as described elsewhere [34], and Ion Torrent S5 XL platform (Thermo Fisher Scientific, Waltham, MA, USA) at The Public Health Agency of Sweden as described elsewhere [35]. The Illumina sequencing reads underwent de novo assembly using SKESA (version 2.3.0), where the reads were assembled into longer contiguous sequences to rebuild an approximate sequence of the original genome [33]. The Ion Torrent sequencing reads were de novo assembled utilizing SPAdes (version 3.12.0) in its “careful mode”, a specialized setting designed to enhance the accuracy, comprehensive coverage, and fidelity of the assembly process, resulting in a more reliable reconstruction of the genomic sequence, and then the sequences were annotated with Prokka (version 1.14.6) [33]. The genomic assemblies in this study were deposited in GenBank with accession numbers, as shown in Appendix A.

Serotype determination was achieved by comparing assemblies to the SerotypeFinder database (DTU, Denmark) (http://www.genomicepidemiology.org/ (accessed on 6 August 2020)) with the use of BLAST+ (version 2.2.30) [33]. An in-house *stx* subtyping database was constructed with ABRicae (version 0.8.10) (https://github.com/tseemann/abricate (accessed on 6 August 2020)), incorporating representative nucleotide sequences of all identified *stx1* and *stx2* subtypes, and then *stx* subtypes were identified using the assemblies to search against this *stx* subtyping database. The presence of intimin-encoding gene *eae* was determined according to the genome annotation as previously described [33]. Multi-locus sequence typing (MLST) analysis was performed by comparing sequences of seven housekeeping genes (*adk*, *fumC*, *gyrB*, *icdF*, *mdh*, *purA*, and *recA*) against the *E. coli* MLST database with the use of an online tool provided by the Warwick *E. coli* MLST scheme website (https://enterobase.warwick.ac.uk/species/ecoli/allele_st_search (accessed on 8 August 2020)) as mentioned before [33]. The allelic profile of these seven housekeeping genes was used to generate a specific sequence type (ST) for each STEC strain. The metadata of all isolates is shown in Appendix A.

### 2.3. espP Subtyping

The sequences of the *espP* gene were retrieved from the genomic assemblies in accordance with the genome annotation. The unique *espP* sequences in this study were then aligned with reference nucleotide sequences of different *espP* subtypes that have been previously reported and downloaded from GenBank [22,36,37]. After alignment using MEGA 11 software (version 11.0.13) (Center for Evolutionary Medicine and Informatics, Tempe, AZ, USA), the genetic distances of the *espP* sequences were calculated with the maximum composite likelihood method, and a neighbor-joining phylogenetic tree was constructed using 1000 bootstrap replicates with maximum composite likelihood model. The *espP* subtypes were determined by the phylogenetic structure and genetic distance. Based on *espP* sequence polymorphism, *espP* genotypes (GTs) were used to determine the diversity within each *espP* subtype as described previously [38]. 

### 2.4. Data Analysis

Statistical correlations between the presence of *espP*/*espP* subtypes and characteristics of the strain (serogroups, *stx* subtypes, the presence of *eae*) or clinical outcomes (HUS, BD, and NBS) were examined using Fisher’s exact test in R software (version 4.3.1) (https://www.r-project.org) (accessed on 20 November 2023). A *p*-value less than 0.05 was considered statistically significant.

## 3. Results

### 3.1. Incidence of espP in STEC Strains 

Among 239 clinical STEC strains, *espP* was present in 130 (54.4%), including 45 strains from HUS patients and 85 from non-HUS patients, of which 38 were from patients with BD and 47 from STEC-infected individuals with NBS. A total of 55 serotypes were identified in 239 STEC clinical isolates; 64 out of 65 O157:H7 strains (98.5%) and 66 out of 174 non-O157 strains (37.9%) carried *espP*. Among “top-six” non-O157 serogroups (O26, O45, O103, O111, O121, and O145), 31 out of 38 strains of serotype O26:H11 (81.6%), 18 of 26 O121:H19 (69.2%), 2 of 3 O145:H28 (66.7%), and 1 of 19 O103:H2 strains (5.3%) were positive for *espP*. All O103:H8 and O111:H8 strains were negative for *espP*. In addition, all strains of the remaining non-O157 serotypes (O165:H25, O177:H25, O55:H12, O115:H11, O15:H16, O180:H2, O84:H2, and O98:H21) contained *espP* (Appendix A). In total, 13 serotypes were identified among 130 *espP*-positive STEC strains, with O157:H7 (64) being the most predominant, followed by O26:H11 (31) and O121:H19 (18). Moreover, *espP* was detected in 124 (71.7%) *eae*-positive strains and 6 (9.1%) *eae*-negative strains.

The presence of *espP* was significantly associated with BD, HUS, O157:H7, and the presence of *eae* (Table 1). However, no association was found between *espP* and age groups, nor was the duration of bacterial shedding (Appendix A).

### 3.2. Diversity of espP Subtypes

In total, 130 *espP* sequences were extracted from genomes of *espP*-positive STEC strains, and 18 unique *espP* sequences were determined. Four *espP* subtypes, i.e., α, δ, γ, and ε, were assigned based on phylogenetic structure (Figure 1). GTs were identified in each *espP* subtype to illustrate sequence polymorphisms and represent the diversity within a subtype. Each of the four *espP* subtypes contained 2 to 10 GTs. *espP*α had 10 GTs (GT1–GT10), followed by *espP*γ (GT1–GT3), *espP*δ (GT1–GT3), and *espP*ε (GT1–GT2) (Figure 1). Using BLASTn search against the GenBank database (nr/nt), 15 out of 18 *espP* GTs were found identical to publicly available *espP* sequences, while 3 GTs (α/GT9, α/GT10, and γ/GT1) showed a nucleotide identity ranging from 99.95% to 99.97% with the publicly available *espP* sequences in the database.

Among 130 *espP*-positive strains, *espP*α was the most predominant subtype, present in 119 strains (91.5%). *espP*δ, *espP*γ, and *espP*ε were found in five (3.9%), four (3.1%), and two strains (1.5%), respectively. *espP*β was not detected in our strain collection (Appendix A). Out of the 45 *espP*-positive strains from patients with HUS, 42 (93.3%) harbored subtype α, while 2 (4.5%) harbored δ, and 1 (2.2%) harbored γ. Among the 38 *espP*-positive strains from patients with BD, 37 (97.4%) carried subtype α, and 1 (2.6%) carried subtype δ. However, no association was found between *espP* subtypes and clinical outcomes. No association between *espP* subtypes and the duration of bacterial shedding or age was observed either (Appendix A).

Among 18 *espP* GTs, 2 major GTs (*espP*α/GT2 and *espP*α/GT1) contained 52 and 49 strains, respectively, and 9 GTs contained only 1 strain, while the rest contained 2 to 5 strains (Figure 1). *espP*α/GT2 was more common in strains from patients with BD, HUS, and BD + HUS, whereas *espP*α/GT1 was more prevalent in strains from individuals with NBS and non-HUS, and *espP*α/GT6 was more prevalent in strains with NBS (Table 2). No association was found between other *espP* subtypes/GTs and clinical symptoms (Appendix A).

### 3.3. Correlation between espP Subtypes and Serotypes

In total, 130 *espP*-positive strains were classified into 13 O:H serotypes. The most predominant serotype was O157:H7 (64/130, 49.2%), followed by O26:H11 (31/130, 23.9%) and O121:H19 (18/130, 13.9%). *espP*α was present in all O157:H7 (64), O26:H11(31), O121:H19 (18), O177:H25 (3), O145:H28 (2), and O103:H2 (1) strains and statistically associated with O157:H7 (Appendix A), while *espP*γ, *espP*δ, and *espP*ε were found in non-O157 serogroups, among which *espP*γ was associated with O55:H12, O98:H21, O180:H2, and O84:H2 serotypes, *espP*δ was associated with O165:H25, and *espP*ε was associated with O115:H10 and O15:H16 (Appendix A). 

A correlation was observed between serotypes and *espP* GTs. Each *espP* genotype contained one, two, or four different serotypes, while each serotype was designated to one *espP* genotype with the exception of serotypes O165:H25, O121:H9, O26:H21, and O157:H7 (Figure 1). O165:H25 strains were assigned to *espP* genotypes δ/GT1, δ/GT2, or δ/GT3, O121:H19 strains were assigned to α/GT1, α/GT8, or α/GT9, O26:H11 strains were assigned to α/GT1 and α/GT10, while O157:H7 strains carried α/GT2, α/GT3, α/GT4, or α/GT5 (Figure 1).

### 3.4. Distribution of stx/stx Subtypes in espP-Positive Strains

Among 130 *espP*-positive STEC strains, 33 strains contained *stx1* only, 85 strains carried *stx2* only, and 12 strains harbored both *stx1* and *stx2*. *stx2* (65.4%) was more prevalent than *stx1* (25.4%). One *stx1* subtype (*stx1a*) and three *stx2* subtypes (*stx2a*, *stx2c,* and *stx2g*) were detected. A total of seven *stx* subtypes and combinations were identified, namely, *stx2a* + *stx2c*, *stx1a*, *stx2a*, *stx2c*, *stx1a* + *stx2c*, *stx1a* + *stx2a*, and *stx2g*. *stx2a* + *stx2c* (36.9%) was the most predominant, followed by *stx1a* (25.4%) and *stx2a* (19.2%). Of 119 *espP*α-containing strains, *stx2a* + *stx2c* (45) was the most predominant subtype, followed by *stx1a* (28), *stx2a* (23), and *stx2c* (11) (Table 3). Among five STEC strains with *espP*γ, four strains contained *stx1a,* and one strain carried *stx2a*. Four *espP*δ-positive STEC strains carried *stx2a* + *stx2c* and *st*x*2a*. Two *espP*ε-positive strains harbored *stx1a* and *stx2g* (Figure 1). *stx2c*, *stx1a* + *stx2c,* and *stx1a* + *stx2a* subtypes were only present in strains carrying *espP*α, while *stx2g* was only found in strains possessing *espP*ε. *espP*γ and *espP*ε were more prevalent in strains with *stx1a* and *stx2g* (*p* = 0.0147 and 0.0154), respectively, whereas no association was found between *espP*α and *stx* subtypes (Appendix A). 

Combinations of *stx* subtypes and *espP* subtypes showed associations with clinical symptoms. The presence of *stx2a* + *stx2c* + *espP*α was significantly higher in strains from patients with HUS and BD + HUS (*p* < 0.0001), while strains with *stx1a* + *espP*α were more prevalent in patients without HUS or BD + HUS (*p* = 0.0003 and 0.0036). Additionally, *stx1a + stx2c* + *espP*α was more commonly found in strains associated with non-HUS (*p* = 0.0500), and *stx2c* + *espP*α showed a higher prevalence in strains with NBS (*p* = 0.0383 and 0.0172) (Table 3). 

## 4. Discussion

This study reported a high prevalence of *espP* (54.4%) in clinical STEC strains from patients with various disease outcomes, especially in strains of O157 (98.5%), O26 (81.6%), O121 (69.2%) and O145 (66.7%) serogroups, and *espP* was detected in 75.0% of strains from patients with HUS, 74.5% of strains with BD and 74.8% of strains with BD + HUS. The prevalence and distribution of the *espP* gene in human-derived STEC strains have also been investigated in previous studies. For instance, *espP* was detected in 55.0% of STEC strains implicated in human disease in Africa [39], whereas 65.0% of clinical STEC strains harbored *espP* in Austria, with the majority being serogroup O157, O26, and O145 strains [28]. *espP* was observed in the majority of O145 STEC strains (88.0%) from patients with watery diarrhea, BD HUS, and from Germany [40]. In a report from Canada, *espP* was present in 86 (76.8%) out of 112 STEC strains of highly pathogenic serogroups O157, O26, O103, O111, and O145 from humans, including 42 (77.8%) strains from patients with severe diseases (BD + HUS) [41]. Meanwhile, *espP* was not detected in *stx1c*- and *stx2e*-harbouring *eae*-negative STEC isolates from patients in this study, the same as previously described [42,43]. These findings showed that *espP* tended to be prevalent in clinical STEC strains, especially in highly pathogenic serogroups, suggesting its role in the pathogenic process and clinical outcomes. 

There are very limited data on the association of *espP* and disease severity, although an antibody response against EspP was discovered during the development of STEC infection [18]. We found that *espP* was strongly associated with severe outcomes, e.g., BD and/or HUS, in contrast with a previous study in Canada reporting no significant association between EspP protease and disease in humans [41]. We were interested to understand if different *espP* subtypes contribute to the disease severity. A previous study revealed significant functional differences among various EspP subtypes, where subtype α and γ isolates showed proteolytic activity, whereas subtype β and δ either lacked proteolytically activity or were not secreted, and these differences correlated with point mutations around the active serine protease site [22]. Subtype ε was first found in O91:H14 strains, with no functional study till now [37]. Four *espP* subtypes, i.e., *espP*α, *espP*γ, *espP*δ, and *espP*ε were identified in our strain collection, in which *espP*α was the most predominant subtype, accounting for 97.5% of strains belonging to serotypes O157:H7, O26:H11, O121:H19, O103:H2, and O145:H28, while other serotypes harbored *espP*γ, *espP*δ or *espP*ε. *espP*α was the predominant subtype, especially in strains from patients with BD and HUS, and was statistically associated with O157:H7. Although no association was found between four *espP* subtypes and clinical outcomes, within ten *espP*α genotypes, *espP*α/GT2 was significantly associated with BD, HUS, and BD + HUS, as compared to other *espP*α genotypes, indicating that certain *espP*α genotypes could be considered as a predictor for severe disease outcome. Further studies are necessary to understand the functional differences and mechanisms of different *espP*α genotypes underlying STEC pathogenesis. 

The coexistence of *stx* and other virulence genes, i.e., *eae*, is more prone to enhance the virulence of STEC and exacerbate the STEC-associated disease severity [44]. However, there is limited literature describing the relationship between the coexistence of *stx* subtypes and *espP* in relation to disease severity. Our study showed that *stx2a* + *stx2c* was the most prevalent *stx* subtype among *espP*-positive clinical strains. Interestingly, the presence of *stx2a* + *stx2c* + *espP*α (mostly *espP*α/GT2) was strongly associated with BD and HUS, indicating that *espP*α might play a more important role in the pathogenesis of STEC strains with *stx2a* + *stx2c*. In accordance with a previous study showing that 97 out of 106 *espP*-positive strains (91.5%) from humans were positive for *eae* [28], 124 out of 130 *espP*-positive strains (95.4%) contained *eae* in this study, and the presence of *espP* was significantly associated with *eae*-positive strains. Intimin encoded by *eae* is an outer membrane protein and responsible for intimate adherence to target eukaryotic cells as an important virulence factor, whereas EspP is an autotransporter that can translocate through the periplasm and the outer membrane of bacteria. The role of EspP in bacterial adhesion was supported by a transposon mutagenesis investigation performed in the O157:H7 STEC strain EDL933, in which EspP was identified as one of the virulence factors directly involved in biofilm formation and adherence to T84 intestinal epithelial cells, probably through the polymerization of EspP and generation of “rope-like structures” [17,30]. It has also been demonstrated that human STEC isolates that carry *eae* along with *espP* adhere more strongly to HEp-2 cell cultures [28,41]. Combined with our findings, there might be some functional associations between the two proteins, which need further verification.

In conclusion, *espP* was highly prevalent in clinical STEC strains in Sweden, which was also strongly linked to the presence of the *eae* gene and significantly associated with severe disease outcomes, i.e., BD and HUS. Four *espP* subtypes were identified, among which *espP*α was the most predominant, carried by strains of virulent serogroups O157, O26, O145, O121, and O103, and correlated with serotype O157:H7. *espP*α, along with *stx2a* + *stx2c,* was closely related to HUS, while genotype *espP*α/GT2 was distinctively correlated with BD and HUS, compared to other *espP*α genotypes. Our results revealed that *espP*α, particularly *espP*α/GT2, is prone to be present in highly virulent STEC clinical strains, highlighting its significant clinical relevance. The pathogenicity of *espP*-positive strains associated with human diseases requires further exploration.

## Figures and Tables

**Figure 1 microorganisms-12-00589-f001:**
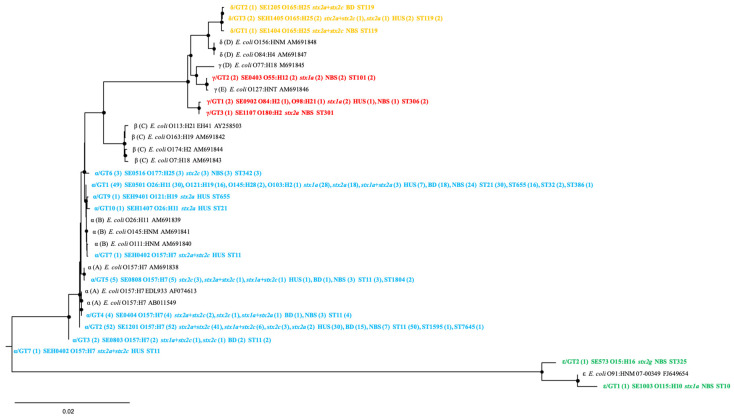
Phylogenetic relationships of 18 different *espP* sequences identified in this study and 15 *espP* subtypes reference sequences based on the Neighbor-Joining method. At each node, the black circles represent values of bootstraps that were more than 60. The corresponding *espP* subtype (number of strains), strain name, serotype (number of strains), *stx* subtype (number of strains), and ST types (number of strains) are shown. The *espP* subtypes/GTs in this study are indicated in bold and different colors. Scale bar indicates genetic distance.

**Table 1 microorganisms-12-00589-t001:** Prevalence of *espP* gene in 239 STEC clinical strains in correlation to clinical symptoms and bacterial features ^#^.

*espP*	Clinical Symptoms									Bacterial Features					
	HUS (*n* = 60)	Non-HUS (*n* = 179)	*p*-Value	BD (*n* = 51)	NBS (*n* = 128)	*p*-Value	BD + HUS (n = 111)	NBS (*n* = 128)	*p*-Value	O157 (*n* = 65)	Non-O157 (*n* = 174)	*p*-Value	*eae*-Positive (*n* = 173)	*eae*-Negative (*n* = 66)	*p*-Value
Positive	45 (75.0)	85 (47.5)	0.0003 *	38 (74.5)	47 (36.7)	<0.0001 *	83 (74.8)	47 (36.7)	<0.0001 *	64 (98.5)	66 (37.9)	<0.0001 *	124 (71.7)	6 (9.1)	<0.0001 *
Negative	15 (25.0)	94 (52.5)		13 (25.5)	81 (63.3)		28 (25.2)	81 (63.3)		1 (1.5)	108 (62.1)		49 (28.3)	60 (90.9)	

HUS—hemolytic uremic syndrome; BD—bloody diarrhea; NBS—non-bloody stool. ^#^ The association was analyzed between *espP* gene and clinical symptoms (non-HUS and HUS; NBS and BD; NBS and BD + HUS), bacterial features (serotype O157 and non-O157; *eae*-positive or *eae*-negative), age groups (child: <10 years; adult: ≥10 years) or duration of bacterial shedding (long: >24 days; short: ≤24 days); only differences with statistical significance were shown. The figures represent the number of *espP*-positive or -negative STEC strains, and the percentage is shown in the following brackets. * Statistically significant difference.

**Table 2 microorganisms-12-00589-t002:** Association between *espP*α/GTs and clinical symptoms ^#^.

*espP*α/GTs	HUS (*n* = 42)	non-HUS (*n* = 77)	*p*-Value	BD (*n* = 37)	NBS (*n* = 40)	*p*-Value	BD + HUS (*n* = 79)	NBS (*n* = 40)	*p*-Value
α/GT1	7 (16.7)	42 (54.5)	<0.0001 *	18 (48.6)	24 (60.0)	0.3647	25 (31.6)	24 (60.0)	0.0054 *
α/GT2	30 (71.4)	22 (28.6)	<0.0001 *	15 (40.5)	7 (17.5)	0.0422 *	45 (57.0)	7 (17.5)	<0.0001 *
α/GT3	0 (0.0)	2 (2.6)	0.5494	2 (5.4)	0 (0.0)	0.2276	2 (2.5)	0 (0.0)	0.5499
α/GT4	0 (0.0)	4 (5.2)	0.2958	1 (2.7)	3 (7.5)	0.6161	1 (1.3)	3 (7.5)	0.1098
α/GT5	1 (2.4)	4 (5.2)	0.6552	1 (2.7)	3 (7.5)	0.6161	2 (2.5)	3 (7.5)	0.3332
α/GT6	0 (0.0)	3 (3.9)	0.5512	0 (0.0)	3 (7.5)	0.2413	0 (0.0)	3 (7.5)	0.0361 *
α/GT7	1 (2.4)	0 (0.0)	0.3529	0 (0.0)	0 (0.0)	1	1 (1.3)	0 (0.0)	1
α/GT8	1 (2.4)	0 (0.0)	0.3529	0 (0.0)	0 (0.0)		1 (1.3)	0 (0.0)	1
α/GT9	1 (2.4)	0 (0.0)	0.3529	0 (0.0)	0 (0.0)	1	1 (1.3)	0 (0.0)	1
α/GT10	1 (2.4)	0 (0.0)	0.3529	0 (0.0)	0 (0.0)	1	1 (1.3)	0 (0.0)	1

NBS—non-bloody stool; BD—bloody diarrhea; HUS—hemolytic uremic syndrome. ^#^ The association was analyzed between *espP*α/GTs and clinical symptoms (HUS and non-HUS; BD and NBS; HUS + BD; and NBS). The number represents the number of strains, and the percentage is shown in the following brackets. * Statistically significant difference.

**Table 3 microorganisms-12-00589-t003:** Association between *stx* subtypes + *espP* subtypes and clinical symptoms. NBS—non-bloody stool; BD—bloody diarrhea; HUS—hemolytic uremic syndrome.

*stx* + *espP*	No. (%)		*p*-Value	No. (%)		*p*-Value	No. (%)		*p*-Value
	HUS (*n* = 45)	non-HUS (*n* = 85)		BD (*n* = 38)	NBS (*n* = 47)		BD + HUS (*n* = 83)	NBS (*n* = 47)	
* **stx1 + espP** *	**3 (6.7)**	**30 (35.3)**	**0.0003 ***	**9 (23.7)**	**21 (44.7)**	**0.0672**	**12 (14.5)**	**21 (44.7)**	**0.0003 ***
*stx1a* + *espP*α	2 (4.4)	26 (30.6)	0.0003 *	9 (23.7)	17 (36.2)	0.2442	11 (13.3)	17 (36.2)	0.0036 *
*stx1a* + *espP*ε	0 (0.0)	1 (1.2)	1	0 (0.0)	1 (2.1)	1	0 (0.0)	1 (2.1)	0.3615
*stx1a* + *espP*γ	1 (2.2)	3 (3.5)	1	0 (0.0)	3 (6.4)	0.2496	1 (1.2)	3 (6.4)	0.1342
* **stx2 + espP** *	**41 (91.1)**	**44 (51.8)**	**<0.0001 ***	**21 (55.3)**	**23 (48.9)**	**0.6636**	**62 (74.7)**	**23 (48.9)**	**0.0040 ***
*stx2a* + *stx2c* + *espP*α	28 (62.2)	17 (20.0)	<0.0001 *	11 (29.0)	6 (12.8)	0.1004	39 (47.0)	6 (12.8)	<0.0001 *
*stx2c* + *espP*α	2 (4.4)	9 (10.6)	0.3281	1 (2.6)	8 (17.0)	0.0383 *	3 (3.6)	8 (17.0)	0.0172 *
*stx2a* + *espP*α	9 (20.0)	14 (16.5)	0.6349	8 (21.1)	6 (12.8)	0.3826	17 (20.5)	6 (12.8)	0.3420
*stx2a* + *espP*δ	1 (2.2)	0 (0.0)	0.3462	0 (0.0)	0 (0.0)	1	1 (1.2)	0 (0.0)	1
*stx2a* + *stx2c* + *espP*δ	1 (2.2)	2 (2.4)	1	1 (2.6)	1 (2.1)	1	2 (2.4)	1 (2.1)	1
*stx2a* + *espP*γ	0 (0.0)	1 (1.2)	1	0 (0.0)	1 (2.1)	1	0 (0.0)	1 (2.1)	0.3615
***stx1* + *stx2* + *espP***	**1 (2.2)**	**11 (12.9)**	**0.0565**	**8 (21.1)**	**3 (6.4)**	**0.0566**	**9 (10.8)**	**3 (6.4)**	**0.5349**
*stx1a* + *stx2c* + *espP*α	0 (0.0)	8 (9.4)	0.0500 *	6 (15.8)	2 (4.3)	0.1315	6 (7.2)	2 (4.3)	0.7101
*stx1a* + *stx2a* + *espP*α	1 (2.2)	3 (3.5)	1	2 (5.3)	1 (2.1)	0.5841	3 (3.6)	1 (2.1)	1
*stx2g* + *espP*ε	0 (0.0)	1 (1.2)	1	0 (0.0)	1 (2.1)	1	0 (0.0)	1 (2.1)	0.3615

* Statistically significant difference.

## Data Availability

The genome assemblies of all strains in this study were deposited in GenBank with accession numbers and metadata shown in Appendix A.

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
