# Peer review of "Prevalence and Characteristics of Plasmid-Encoded Serine Protease EspP in Clinical Shiga Toxin-Producing Escherichia coli Strains from Patients in Sweden"

_microorganisms, 2024, doi:10.3390/microorganisms12030589_

Round 1

Reviewer 1 Report

Comments and Suggestions for Authors

I think that the manuscript entitled “Prevalence and Characteristics of Plasmid-Encoded Serine Protease EspP in Clinical Shiga Toxin-Producing Escherichia coli strains from Patients in Sweden” is in principle suited for a publication in Microorganisms, Special Issue “Research on Foodborne Pathogens and Disease”. The authors systematically analyze the distribution of espP subtypes, their correlation with serotypes, stx subtypes, and clinical symptoms. They also investigate the association between espP and other virulence factors, such as the eae gene, shedding light on potential mechanisms underlying STEC pathogenesis. The article contributes to the field by providing insights into the prevalence and diversity of espP subtypes among clinical STEC strains, as well as their association with disease severity. The authors use appropriate statistical analyses to support their findings and provide comprehensive tables and figure to illustrate key points. However, I have some minor comments regarding the presentation of the methodologies used in the study.

Comments:

Lines 115-116. “The isolation of STEC strains was performed as described previously [32].” Unfortunately, in reference #32, I did not find a detailed description of the isolation of STEC strains. Please provide the corresponding methodology used in this manuscript. This would be helpful and convenient for readers, especially if the authors made their own modifications to the isolation methodology.

Lines 121-124. “…as briefly described [33]”. Additionally, the referenced work #33 does not provide specific information on sample preparation procedures and DNA sequencing protocol. Please provide a link to the article where this is described or describe it in this article (as supplementary materials to the article).

Line 151. "GeneBank" should be correct to "GenBank".

Author Response

Comments:

Lines 115-116. “The isolation of STEC strains was performed as described previously [32].” Unfortunately, in reference #32, I did not find a detailed description of the isolation of STEC strains. Please provide the corresponding methodology used in this manuscript. This would be helpful and convenient for readers, especially if the authors made their own modifications to the isolation methodology.

We have replaced this reference  with the correct one where the isolation of STEC strains was described in detail. We thanks for the correction.

Lines 121-124. “…as briefly described [33]”. Additionally, the referenced work #33 does not provide specific information on sample preparation procedures and DNA sequencing protocol. Please provide a link to the article where this is described or describe it in this article (as supplementary materials to the article).

We thank for the reviewer´s good comment.

We have now cited specifically the orignal two publications where the sample preparations and DNA sequencing protocols on Illumina HiSeq X platform and Ion Torrent S5 XL platform were described in detail.

Line 151. "GeneBank" should be correct to "GenBank".

Thank for the correction. It is corrected.

Reviewer 2 Report

Comments and Suggestions for Authors

The topic discussed is of great interest, the introduction includes relevant references and provides a sufficient overview of the topic at hand. The bibliographic references are relevant for the research. The experimental design is appropriate. The methods are adequately described and the results are clear and not misleading.Overall, the study provides a detailed overview of the genetic diversity of STEC strains. The results show that 130 espP sequences were extracted from the genomes of 194 espP-positive STEC strains, with eighteen unique espP sequences identified. The four identified espP subtypes are α, δ, γ, and ε, each containing 2 to 10 Genotypes (GT). It is interesting to note that some espP GT sequences are identical to publicly available ones, while others show slight variations in nucleotide identity. The espPα subtype was found to be the most predominant, present in approximately 91.5% of strains, followed by espP δ, γ, and ε. Variations in the presence of espPα, δ, and γ subtypes were observed among strains from patients with different disorders. No significant associations were found between espP subtypes and clinical outcomes. Furthermore, a correlation was identified between espP subtypes and serotypes, with certain serotypes associated with specific espP subtypes. A correlation was also observed between serotypes and espP Genotypes. These results provide important information on the genetic diversity and distribution of espP subtypes within the analyzed STEC strains. Among the 130 espP-positive STEC strains, various gene combinations were identified. Specifically, seven subtypes and combinations of stx genes were found, including stx1a, stx2a, stx2c, stx2g, and various combinations of them. Among STEC strains with α espP, the predominant subtype was found to be stx2a+stx2c. Associations were found between specific combinations of stx subtypes and clinical symptoms. For example, the presence of stx2a+stx2c+espPα was significantly higher in strains associated with certain symptoms compared to others. These data provide important insights into the distribution of STEC strains and their correlation with clinical symptoms.The text extensively discusses the high prevalence of the espP gene in clinical STEC strains across different serogroups and its association with severe disease outcomes such as HUS and HC. The importance of the presence of espP in combination with other virulence genes, such as eae, in increasing the virulence of STEC is highlighted. The results regarding different espP subtypes and their association with disease severity are presented, suggesting that certain espPα genotypes may be predictors of more severe clinical outcomes. Functional associations between EspP and Eae are also discussed, indicating the need for further investigation. Finally, the importance of additional studies to better understand the role of different espP genotypes in the pathogenesis of STEC-associated human diseases is emphasized.The conclusions are supported by the results.

Author Response

Thank you for nice comments to our manuscript describing the main findings.